# Anti-IgG Doped Melanin Nanoparticles Functionalized Quartz Tuning Fork Immunosensors for Immunoglobulin G Detection: In Vitro and In Silico Study

**DOI:** 10.3390/s24134319

**Published:** 2024-07-03

**Authors:** Dilhan Gürcan, Engin Baysoy, Gizem Kaleli-Can

**Affiliations:** 1Department of Biomedical Engineering, İzmir Democracy University, İzmir 35140, Türkiye; 2Department of Biomedical Engineering, Bahçeşehir University, İstanbul 34353, Türkiye

**Keywords:** immunosensor, quartz tuning fork, melanin nanoparticles, simulation modeling, immunoglobulin G detection, biosensor design optimization

## Abstract

The quartz tuning fork (QTF) is a promising instrument for biosensor applications due to its advanced properties such as high sensitivity to physical quantities, cost-effectiveness, frequency stability, and high-quality factor. Nevertheless, the fork’s small size and difficulty in modifying the prongs’ surfaces limit its wide use in experimental research. Our study presents the development of a QTF immunosensor composed of three active layers: biocompatible natural melanin nanoparticles (MNPs), glutaraldehyde (GLU), and anti-IgG layers, for the detection of immunoglobulin G (IgG). Frequency shifts of QTFs after MNP functionalization, GLU activation, and anti-IgG immobilization were measured with an Asensis QTF F-master device. Using QTF immunosensors that had been modified under optimum conditions, the performance of QTF immunosensors for IgG detection was evaluated. Accordingly, a finite element method (FEM)-based model was produced using the COMSOL Multiphysics software program (COMSOL License No. 2102058) to simulate the effect of deposited layers on the QTF resonance frequency. The experimental results, which demonstrated shifts in frequency with each layer during QTF surface functionalization, corroborated the simulation model predictions. A modelling error of 0.05% was observed for the MNP-functionalized QTF biosensor compared to experimental findings. This study validated a simulation model that demonstrates the advantages of a simulation-based approach to optimize QTF biosensors, thereby reducing the need for extensive laboratory work.

## 1. Introduction

Immunoglobulin G (IgG) plays a pivotal role in the immune response, serving as a primary defense mechanism against pathogens [1,2]. Detecting IgG levels can provide valuable information about an individual’s immune status, exposure to infections, and potential immunity against specific diseases [3,4,5]. Regarding infectious diseases such as COVID-19, the detection of IgG plays a key role in understanding the immune response post-infection and assessing the development of immunity over time [6]. Recent studies suggest that the presence of anti-SARS-CoV-2 IgG can indicate past exposure to the virus and potentially confer protection against reinfection [3]. As a result of monitoring the levels of IgG in COVID-19 patients as well as in healthy individuals, it may be possible to ascertain the duration of immunity and to guide public health strategies, including vaccination campaigns and infection control measures [6]. IgG is also vital for diagnosing and treating autoimmune disorders and thrombotic conditions. A significant proportion of patients with thrombotic microangiopathies and severe ADAMTS13 deficiency have anti-ADAMTS13 IgG, highlighting the usefulness of IgG detection in these clinical settings [7]. The identification of anti-mitochondrial IgG antibodies has been linked to specific signs and symptoms of systemic lupus erythematosus, emphasizing the importance of detecting IgG antibodies for understanding the pathogenesis of the disease [8]. As a result of the advancement of biosensor technologies, sensitive and rapid methods of detecting IgG have been developed, which are facilitating point-of-care testing and personalized medicine practices [9]. Over the last few decades, innovative biosensors have revolutionized diagnostic capabilities for IgG detection, enabling early detection of diseases and monitoring of therapeutic outcomes. In addition to standard analytical methods, such as the enzyme-linked immunosorbent assay (ELISA), there are several biosensor designs that use chemical, electrochemical, optical, and mass sensitive techniques to measure IgG levels in blood and plasma [1,10,11,12,13,14,15,16,17]. Among these approaches, quartz crystal microbalance (QCM) biosensors are one of the most preferred mass-sensitive biosensors and have been extensively studied for the detection of IgG by monitoring frequency shifts following IgG binding to a crystal [18,19,20,21]. Compared to QCM-based biosensors, quartz tuning fork (QTF) biosensors possess distinct advantages, including high quality factor, sensitivity, repeatability, frequency stability, sharp frequency response, and low cost [22,23,24,25,26,27,28,29,30,31]. These sensors can detect biomolecules in liquid environments by converting the mass of the target analytes into a signal, represented as the resonance frequency of the QTF using an oscillator circuit and frequency counter [27,31,32,33,34,35,36,37,38]. The performance of the sensor is influenced by the interaction between the target biomolecule and the surface of the resonator. The modification of the transducer’s surface through the creation of a recognition layer is essential due to the chemical inertness of the transducer’s surface, enhancing the sensitivity and selectivity of mass-sensitive biosensors for biomolecule detection [31]. Due to their small size, surface modifications of QTFs are challenging and require the use of specialized equipment, materials, reagents, and numerous repetitive laboratory tests. To address these challenges, several simulation-based studies have been conducted to minimize optimization time and reduce the overall cost of the experimental tests. Atabaki et al. developed a finite element method (FEM)-based model using the COMSOL Multiphysics simulation program for early detection of acute myocardial infarction [39]. Another study simulated the response of polystyrene and polymethylmethacrylate modified QTF surfaces to volatile organic compounds (VOCs) with 97% accuracy [32]. Additionally, Sampson et al. designed graphene nanoribbons, polyaniline and gold nanoparticles capped with a polyvinyl pyrrolidone modified QTF biosensor, to identify VOCs such as ethanol, methanol, chloroform, and acetone with the FEM using COMSOL Multiphysics [33]. A correlation was found between the simulation results and the experimental results. The impressive predictive accuracy of simulations in biosensors opens an exciting opportunity to improve the effectiveness of QTF-based biosensors in biomedical applications via the exploration of innovative nanomaterials through simulation studies.

Melanin nanoparticles (MNPs) have recently gained attention in biosensors and bioelectronic applications as an eco-friendly alternative to metallic and metal oxide nanoparticles [40]. Several sources, including human hair, bacteria, fungi, black tea leaves, insects, chestnut shell, catfish, and cuttlefish are known to produce natural MNPs [41,42,43]. Unlike other sources requiring multiple extraction or purification steps, MNPs can be readily obtained from cuttlefish (*Sepia officinalis*) ink through a centrifugation and washing process. In recent years, MNPs have been extensively used in various research areas due to their remarkable properties, including high biocompatibility, biodegradability, stability, and strong binding ability to biomolecules and metal ions [40,43,44,45,46,47,48,49,50]. Since MNP surfaces contain a wide range of functional groups, especially amine groups, they offer an ideal site for the covalent bonding of bioreceptors onto sensor surfaces [51,52,53,54]. Furthermore, the catechol groups in MNPs can enhance their adhesion to sensor surfaces, contributing to excellent long-term stability. MNPs are therefore an excellent candidate for use as a nanomaterial during the functionalization of biosensors. However, further exploration through simulation studies is needed to understand comprehensively their integration into QTF-based biosensor research.

This study focuses on the development of a QTF immunosensor for the detection of IgG, employing MNPs, GLU, and anti-IgG. Initially, experimental studies determined the resonance frequency changes of the QTF sensor following MNP functionalization, GLU, anti-IgG doping, and IgG detection. Furthermore, an FEM-based QTF model was created to assist in the development of the QTF immunosensor, incorporating both the physical and functional characteristics of QTF into the COMSOL Multiphysics program. Finally, the impact of MNP functionalization, GLU, anti-IgG immobilization, and IgG detection on QTF surfaces with varying thin film thicknesses was investigated in terms of resonance frequency shifts correlating with mass increases. The functionality and accuracy of the developed QTF simulation model were then compared with the experimental results obtained from QTF.

## 2. Methods

### 2.1. Experimental

#### 2.1.1. MNPs Extraction Procedure

MNPs were extracted from commercially available cuttlefish (*Sepia officinalis*) ink sac. Initially, the ink sac was dissected to collect the ink, which was then diluted five-fold with deionized water (DIW). The ink suspension was centrifuged at 10,000 rpm for 20 min. After discarding the supernatant, the pellets were washed with DIW. Salt and impurities were eliminated by repeating this washing and centrifuge procedure three times. Subsequently, the pellets were dried in an oven at 50 °C for 2 days. Just prior to application, a stock solution of MNPs (1 wt% in DIW) was prepared and sonicated overnight using a bath-type sonicator [51].

#### 2.1.2. Surface Modification of QTFs

Prior to surface functionalization, the hermetic case of the QTFs was removed. Each substrate was then cleaned by sonication in ethanol for 30 min. After carefully washing with DIW, the forks were gently dried with nitrogen gas. A dip coating procedure was performed using MNP solution to modify the fork tips at room temperature for 60 min (Figure 1). The change in frequency (Δ*f* = *f*_BASE_ − *f*_MNP_) was calculated by measuring the resonance frequencies of QTFs before (*f*_BASE_) and after MNP coating (*f*_MNP_) using the Asensis QTF F-master device (Asensis, Ankara, Türkiye). 

In the second step, surface activation was performed using a GLU solution (Sigma-Aldrich, St. Louis, MO, USA), a highly reactive dialdehyde reagent widely used as a cross-linker in biosensor applications [30,55]. The amino groups on MNPs are coupled with an aldehyde group of GLU, while the cross-linker is strongly attached to the forks (Figure 1). During the activation, QTFs modified with MNPs (MNP-QTF) were dipped in a 25% aqueous GLU solution (*v:v* in DIW) at room temperature for 2 h. Unbound GLU was removed by washing the forks in DIW three times. The frequency shift (Δ*f* = *f*_MNP_ − *f*_GLU_) was calculated by subtracting the resonance frequencies of GLU activated MNP-QTFs from the resonance frequencies of MNP-QTFs using the Asensis QTF F-master device. The frequency change of each QTF and a statistical analysis of overall change were conducted separately.

Lastly, anti-IgG immobilization, essential for selective IgG detection, was provided. The aldehyde group remaining on the other end of GLU is expected to interact with a bioreceptor, in our case, anti-IgG. First, the optimum concentration of anti-IgG for IgG detection was determined by interacting activated QTF surfaces with different concentrations of anti-IgG (0.5, 1.0, 1.5, 2.0, 2.5, 3.0, and 3.5 µg/mL) (EMD Millipore, Merck KGaA, Darmstadt, Germany) at 4 °C (Figure 1). Then, each sample (*f*_anti-IgG_) was dipped into 15 mg/mL of IgG solution (Sigma, St. Louis, MO, USA) and the frequency shifts were recorded (Δ*f* = *f*_IgG_ − *f*_anti-IgG_). The concentration at which the frequency change stabilized was identified as the point where all active sites interacted with anti-IgG.

#### 2.1.3. Analytical Response of Anti-IgG Immobilized QTFs to IgG

The analytical performance of QTF-based mass sensitive biosensors was tested in different concentrations of IgG (0.1 µg/mL–15 µg/mL in PBS pH: 7.4) solutions at 25 °C and the corresponding frequency shifts were recorded (Δ*f* = *f*_anti-IgG_ − *f*_IgG_). Unmodified QTFs were used as a negative control. 

#### 2.1.4. Characterization

The hydrodynamic size distribution of MNPs was analyzed using dynamic light scattering (DLS) measurements (ZetaSizer NanoZS 90, Malvern, UK). Scanning electron microscopy (SEM) was used to examine the morphology of MNP, unmodified QTF, MNP modified QTFs, and anti-IgG doped MNP modified QTF. Samples were fixed onto aluminum stub surfaces using double-sided carbon tape and coated with a gold–palladium (Au-Pd; 60–40%) mixture via sputter-coating. The specimens were then observed under SEM operating at 7.5 kV, and 30 kV, with an original magnification of 20,000× and 65,000× (Apreo S model-FEG, Thermo Scientific, Waltham, MA, USA). Utilizing ImageJ^®^ software version 1.54d (NIH, Bethesda, MD, USA), the average diameter of the MNPs was determined from SEM images captured at various locations. The results are presented as the mean value ± standard deviation.

X-ray photoelectron spectroscopy (XPS) was used to characterize the chemical composition of MNP modified QTF, anti-IgG loaded MNP modified QTF, and after IgG detection with anti-IgG loaded MNP modified QTF. A Thermo K-Alpha (Thermo Fisher Sci., Waltham, MA, USA) instrument and a Thermo K-Alpha monochromatic high-performance XPS spectrometer (Thermo Fisher Sci., Waltham, MA, USA) at a pressure of 1 × 10^−9^ torr were used for XPS characterization. A 400 μm spot size was used per sample.

### 2.2. Finite Element Method-Based QTF Simulation

A QTF simulation model was set up using the FEM-based COMSOL Multiphysics by following a workflow including the design of the QTF geometry, defining corresponding materials and their characteristics, assigning physics and boundary conditions, creating a mesh, running the model, and postprocessing the results. First, we developed a solid model of the QTF using the piezoelectric crystal (Quartz LH-1978 IEEE), which is predefined in the COMSOL library (Tablo S1).

The QTF dimensions and properties were determined from commercially available QTF used in experimental and previous studies [30,51,53,54,55,56]. Geometric features of the designed QTF model and the thickness of each surface modification are given in Appendix A.

The working principle of QTF biosensors is based on resonance frequency (*f*) shift as a response to change in effective stiffness (*k*: force constant) and additional mass (*m*) on prongs of QTF due to loaded analyte, as shown in the given equation (Equation (1)) below:(1)$$f=\frac{1}{2\pi }\sqrt{\frac{k}{m}}$$

#### 2.2.1. Simulation of Multilayer Thin Film Functionalized QTF

Functional surface modifications are required during the design of QTF based mass-sensitive biosensors. Considering that QTFs are largely composed of quartz and silver, this necessity arises from their inert nature. The forks’ surfaces were activated and functionalized using MNPs, GLU, and anti-IgG, which were layered one over the other, respectively. Therefore, simulation of thin films containing MNPs, GLU, and anti-IgG were applied over QTF prongs, according to the size of each component [53]. The amine groups and catechol present in MNPs are important for biosensor functionality, as well as adhesion to surfaces that result in high stability. After the surface modification of QTFs with MNPs, we expected waviness in the surface topography due to the MNP spherical structure. Since simulating such a complex topography presents some challenges, the coating in the simulation model was assumed to be a thin film. The diameter of this thin film is that of the MNPs, as determined by SEM. Material properties of MNPs, GLU, and anti–IgG containing thin films, predefined in the COMSOL library, are as given in Appendix A. FEM simulation was used to analyze the shifts in resonance frequencies of QTF associated with each thin film-based surface modification.

#### 2.2.2. QTF Immunosensor Modelling

In this section, the modeling implemented to study IgG detection using a multilayer thin film functionalized QTF model was described, based on experimental studies detailed in Section 2. The modeling was performed using COMSOL Multiphysics, with IgG material characteristics defined as shown in Appendix A. The simulation was conducted using a two-dimensional approach rather than a three-dimensional one to simplify the process. Several parameters were defined based on the properties of the QTF and the surface modifications that were applied. A variety of configurations was tested to achieve an immunosensor model, considering both material characteristics and surface modifications.

## 3. Results

### 3.1. QTF Immunosensor

Anti-IgG loaded MNP modified QTF-based mass sensitive biosensors detected IgG by monitoring QTF resonance frequency shifts resulting from mass accumulation in the prongs. First, the inert prongs of QTFs were functionalized with MNPs and activated with a spacer arm and GLU, via the dip coating method. Anti-IgG was then immobilized with GLU, allowing selective interactions with IgG to be achieved. Shifts in the resonance frequency of QTF with respect to additional mass was indicative of IgG detection.

Prior to performing the experiments, resonance frequencies of unmodified QTFs were measured in both air and with IgG, to determine how air damping and matrix affect the frequency. The unsealed QTF resonance frequency was recorded as a base resonance frequency (*f*_BASE_). A decrease in resonance frequency indicated the damping effect of air on the QTF. We observed that after the removal of the hermetic case and cleaning procedures, the resonance frequency was reduced from 32,768 Hz to 32,759 Hz (*n* = 30). Upon dipping the forks in MNP solution (Video S1), the initial frequency shift was found around 17 Hz (Figure 2A) [27]. The activation of amino groups located in MNPs with the GLU caused a significant decrease in resonance frequency (Δ*f = f*_MNP_-*f*_GLU_) with 28.08 ± 9.47 Hz (*n* = 10) (Figure 2B). Zoomed frequency data obtained during MNP immobilization and GLU activation are given in Appendix A.

QTF-based biosensors work on a similar principle to QCM, but the controller used for both surface modification and measurement operates under different forces. Using the Asensis QTF F-master, the frequency shift (Δ*f*) of the biosensor in solution was monitored by recording ten frequency measurements per second with a frequency resolution of 0.01 Hz. The frequency variation in the QTF biosensor positioned in the liquid results from the simultaneous influence of multiple forces. The observed frequency decrease immediately after immersion and its subsequent rise over time can be attributed to the capillary effect of the liquid between the prongs, the greater damping effect of the liquid phase compared to air, the immobilization of MNP, GLU, and anti-IgG on the surface, the binding of IgG to anti-IgG decorated melanin nanoparticles, and temperature-dependent evaporation. Therefore, it is crucial to observe these effects during time-dependent frequency measurements and analyze the moment when the frequency change is recorded.

As shown in Figure 2, when the sensor is immersed in solution, a sharp frequency drop is observed due to the damping effect and capillary effect of the liquid with the immersion. However, owing to the QTF geometry, equilibrium is rapidly achieved, resulting in frequency stabilization. MNP adhesion to the surface is monitored for one hour following immersion. During this process, only a maximum of 40 µm of the prong is in contact with the solution. Over time, the frequency increases due to evaporation.

As the final step of the QTF mass sensitive biosensor design, the optimum antibody concentration for IgG detection was found by monitoring the corresponding frequency shifts with respect to different concentrations of anti-IgG while the IgG concentration was kept constant (Figure 3). For up to 2 µg/mL, the change in resonance frequency gradually increased, whereas no significant change was observed in resonance frequency higher than 2 µg/mL. We observed that the active sites on the surface of the antibody reach saturation at this concentration. Therefore, optimum anti-IgG concentration was determined as 2 µg/mL for our further experiments. Zoomed frequency data obtained during the anti-IgG immobilization and IgG response are given in Appendix A.

### 3.2. Characterization

As depicted in Figure 4a, rough spherical nanoparticles were observed. An agglomeration was observed due to the dominant influence of Van der Waals forces generated by the nano-size. With the help of the ImageJ program, SEM images of the MNPs were analyzed to determine size distributions and the MNP diameter was determined to be 162 ± 20 nm (n = 100). Hydrodynamic size distribution was also conducted for MNPs using the dynamic light scattering technique before the modification steps. After the extraction was completed, the hydrodynamic size of melanin nanoparticles in DI water was determined to be 228 ± 50 nm (n = 3), exhibiting a monodisperse distribution (PDI: 0.12 ± 0.03). Caldas et al. (2020) conducted MNP extraction from cuttlefish ink for controlled release of dexamethasone drug and performed size distribution analysis for its applicability. In this study, the hydrodynamic size of the nanoparticles was reported to be 376.77 ± 62.05 nm, with a PDI of 0.26 ± 0.09 [46].

In Figure 4b, the surface morphology prior to modification of unmodified QTFs is demonstrated, which shows the fork surfaces to be notably smooth. In response to dipping of QTFs in MNP solution (Figure 4c), it was observed that despite localized agglomeration, the surface coating with MNPs was successfully achieved. Based on size distribution, MNPs were found to be located on QTFs with 195 ± 52 nm (n = 100) (Figure 4c). Upon examining the changes in surface morphology and nanoparticle size distributions following interaction with anti-IgG, after the glutaraldehyde activation process, it was observed that both the morphological structure and size remained consistent (189 ± 35 nm (n = 100)) (Figure 4d).

X-ray photoelectron spectroscopy (XPS) was used to analyze MNPs-QTFs, anti-IgG loaded MNPs-QTFs, as well as after IgG detection with anti-IgG loaded MNPs-QTFs (Figure 5). XPS provides detailed insights into surface chemistry with a penetration depth of 10 nm. We analyzed high-resolution XPS spectra of C and N elements. Three components of the C-1s deconvolution peak of MNPs were identified: 287.8 eV, 285.8 eV, and 284.5 eV. A deconvolution peak at 287.8 eV reveals the presence of the carbonyl group (C=O), whereas a peak observed at 285.8 eV indicates the presence of the COC/COH group. The peak at 284.5 eV indicates a C-C bond (Figure 5a_1_). Melanin is primarily composed of eumelanin (98%) which is synthesized enzymatically with 5,6-dihydroxyindole and dihydroxyindole-2-carboxylic acid [57,58]. Thus, the presence of peaks corresponding to the chemical structure of 5,6-dihydroxyindole and dihydroxyindole-2-carboxylic acid in the XPS spectrum indicates that the XPS spectrum successfully identifies the characteristic chemical structure of MNPs. Besides the characteristic MNPs peaks, a new peak formation at 296 eV caused by C-O emerged after loading anti-IgG (Figure 5b_1_). After the IgG detection with QTF-based immunosensor, in addition to the characteristic peaks from anti-IgG loading and MNPs, a new peak was observed at 288 eV, attributed to the -COOH group (Figure 5c_1_). In the N-1s spectrum of melanin nanoparticles, a peak at 399.8 eV is observed that corresponds to C-NH_2_, which is present in the nanoparticle structure (Figure 5a_2_). Anti-IgG loading (Figure 5b_2_) and detection of IgG (Figure 5c_2_) did not result in any significant changes in the peak.

### 3.3. Surface Modification Simulation

After setting up the geometry, corresponding physics, boundary conditions, and mesh construction, the structural mechanics module of COMSOL Multiphysics was used to study the eigenfrequency and frequency domain analysis of QTF. The resonance frequency and vibration mode of QTF were analyzed at room temperature by using a parametric sweep between 0 and 80 kHz. As shown in Figure 6, the peak value was measured in the vibration mode when the eigenfrequency was 32,383 kHz. When compared with the simulation results of the study presented by Ou et al., there is a 0.13% difference in resonance frequency (32,339 kHz) [59]. The shift in frequency is probably caused by the characteristic alteration of selected quartz materials used for the construction of QTF. On the other hand, the resonance frequency of commercially available QTF used in our experimental studies is 32,759 kHz, which refers to a relative error of 1.15% compared to the simulation results.

The prongs of QTF were functionalized with a 210 nm thick layer of MNPs. Consistent with the working principle of QTF as expressed in Equation (1), the additional mass on QTF prongs resulted in a 15-Hz decrease in the resonance frequency of the QTF in simulation. The relative error in the value of the resonance frequency between our previous experimental results [27] and the simulation model is 0.006%. Applying a layer of GLU to the fork surface led to an increase in thickness by 2 nm. It resulted in a 12-Hz reduction in resonance frequency. The effect of the GLU layer on the QTF surface was observed with a 0.05% deviation between the experimental and simulation results. Following chemical activation, anti-IgG was immobilized as a recognition layer on QTF. According to the simulation model, a 10-nm thin film layer of anti-IgG was located on the QTF surface to observe the frequency response of QTF to anti-IgG. Analysis showed that there is a 5-Hz change in frequency caused by the interaction of QTF with anti-IgG. A deviation of 0.026% was observed between the experimental and the simulation data. 

### 3.4. Experimental Biosensing of IgG with QTF Mass Sensitive Biosensor

The analytical performance of the QTF-based immunosensor was evaluated according to frequency shifts at concentrations ranging from 0.1 µg/mL to 15 µg/mL of IgG (Figure 7A). First, unmodified QTFs were tested as a negative control and no frequency change was observed due to the inert nature of prongs. Then, the QTF-based immunosensor was studied and the frequency shifts changed linearly for 1.0 µg/mL to 15.0 µg/mL of IgG, with a regression coefficient of R^2^ = 0.9796 (∆f=(2.3±0.3)C+(23±2.4) (Figure 7B). The current manuscript presents the preliminary findings. These initial results are foundational and form the first steps toward developing robust analytical applications for our QTF-based biosensor. Further research will focus on expanding the range of concentrations tested and optimizing the sensor’s performance to enhance its analytical capabilities.

As first-time researchers began their journey into QTF-based biosensor research, Su et al. evaluated polystyrene coated QTFs immobilized with anti-IgG that operated in the dry phase [38]. Based on this study, the biosensor’s linear range was determined to be between 5 and 100 µg/mL, and beyond 100 µg/mL, the anti-IgG reached saturation. It was seen that modifying the QTFs with anti-IgG-immobilized MNPs reduced the detection limit by up to five-fold compared to polystyrene-coated QTFs.

### 3.5. Simulated QTF Mass Sensitive Biosensor Response 

For the analytical performance of the QTF biosensor, the effect of interaction between anti-IgG and IgG on modified QTF prong surfaces was investigated. For this purpose, the QTF surface was coated with a thin film of IgG with varied thicknesses including 4.5 nm, 8.5 nm, and 12.5 nm. The FEM analysis revealed that the interaction between anti-IgG and IgG caused 14, 24, and 37-Hz shift in resonance frequency of the QTF biosensor, respectively as shown in Figure 8.

Experimental data were consistent with the results obtained from the FEM analysis with a deviation of 0.04%. Moreover, Su et al. found that the selective interaction between anti-lgG and IgG affected the resonance frequency of the QTF biosensor in a similar manner [38].

## 4. Conclusions

In this study, we introduced a melanin nanoparticles (MNPs) functionalized quartz tuning fork (QTF) immunosensor for the detection of immunoglobulin G (IgG), conducting both experimental and simulation studies. The QTF prongs were modified through MNP functionalization, surface activation with glutaraldehyde (GLU), and immobilization of anti-IgG, all aimed at the selective detection of IgG. We determined the corresponding changes in resonance frequency for each modification step through experimental studies. In our simulations, we modeled MNPs, GLU, anti-IgG, and IgG as deposited thin film layers on the QTF, using characteristics identical to those in our experiments. We analyzed the resonance frequency changes for each deposited thin film layer using the finite element method (FEM) based on the COMSOL Multiphysics simulation program. The simulation results showed negligible deviation in resonance frequency compared to the experimental results, demonstrating our model’s accuracy. Notably, while experimental results indicated antibody saturation at a concentration of 2 µg/mL, the simulation did not show any limitations at the active sites on the antibody surface. Although the current QTF simulation model could be further improved for better material characterization and detailed physics considerations, we believe it will significantly aid in determining the optimal parameters for surface modification of QTF biosensors, thereby facilitating future experimental work.

## Figures and Tables

**Figure 1 sensors-24-04319-f001:**
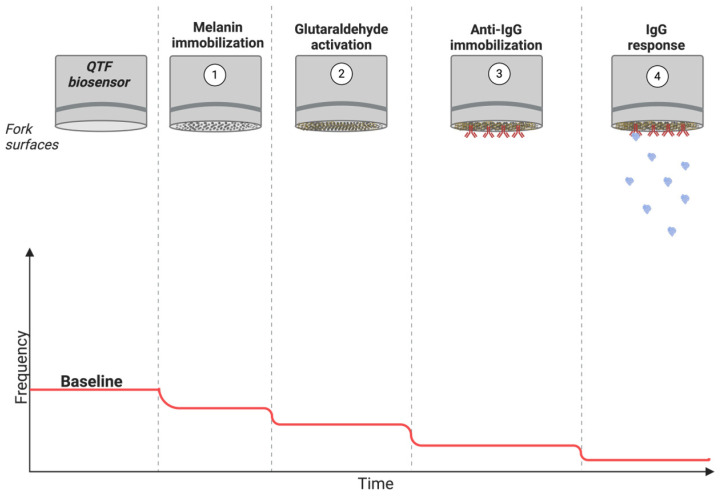
Schematic representation of biosensing of IgG with anti-IgG loaded MNP modified QTF-based biosensor.

**Figure 2 sensors-24-04319-f002:**
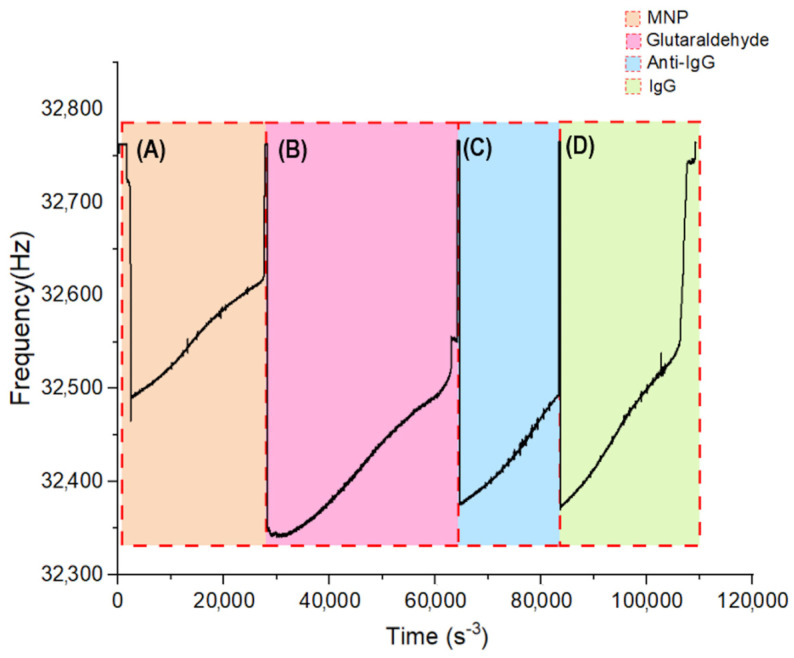
A graph depicting the real-time changes in frequency of QTF collected during surface modification steps, and the real-time responses to IgG. Frequency change during (**A**) MNP coating, (**B**) GLU activation, (**C**) anti-IgG immobilization, and (**D**) anti-IgG-IgG interaction.

**Figure 3 sensors-24-04319-f003:**
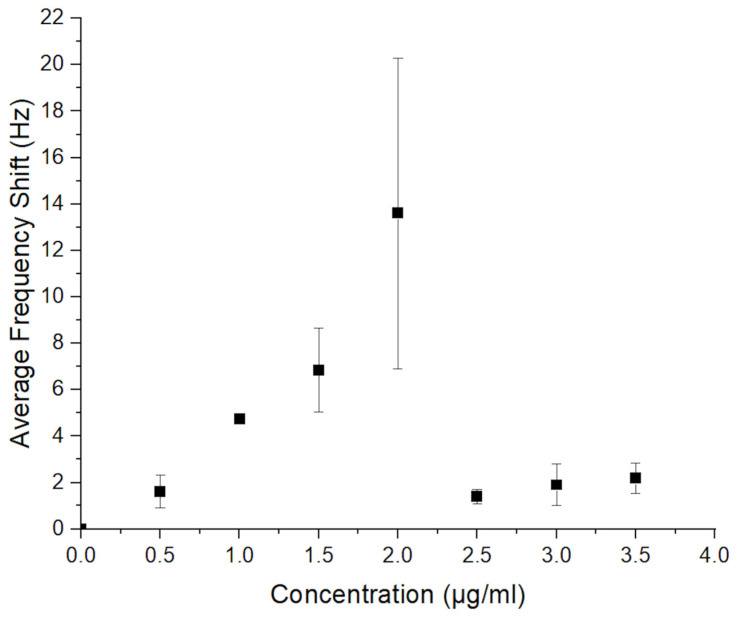
Optimization of the antibody concentrations (*n* = 3).

**Figure 4 sensors-24-04319-f004:**
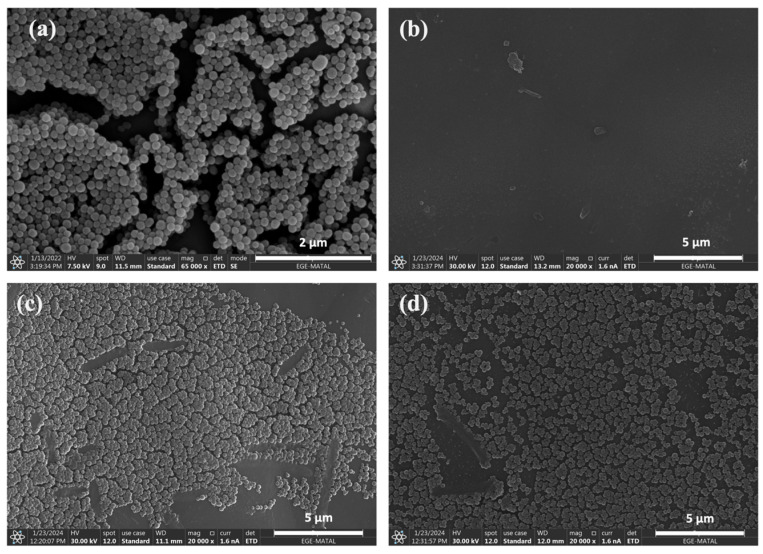
SEM images of (**a**) MNPs, (**b**) unmodified QTF, (**c**) MNPs modified QTF, and (**d**) anti-IgG doped MNPs modified QTF.

**Figure 5 sensors-24-04319-f005:**
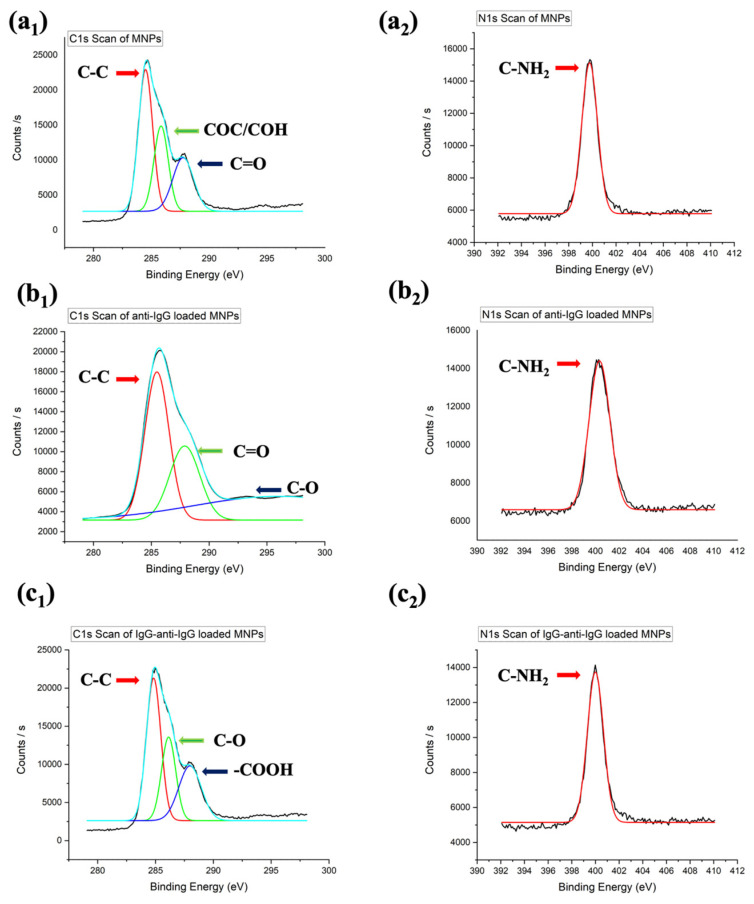
Deconvolution graph of C-content for (**a_1_**) MNPs-QTFs, (**b_1_**) anti-IgG loaded MNPs-QTFs, (**c_1_**) after IgG detection with anti-IgG loaded MNPs-QTFs and N-content for (**a_2_**) MNPs-QTFs, (**b_2_**) anti-IgG loaded MNPs-QTFs, (**c_2_**) after IgG detection with anti-IgG loaded MNPs-QTFs. In the graph, the colors indicate different chemical groups.

**Figure 6 sensors-24-04319-f006:**
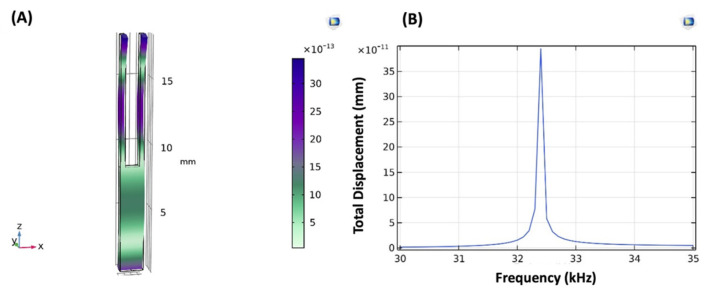
(**A**) Vibration mode of QTF. (**B**) Eigenfrequency of QTF at 32,383 kHz.

**Figure 7 sensors-24-04319-f007:**
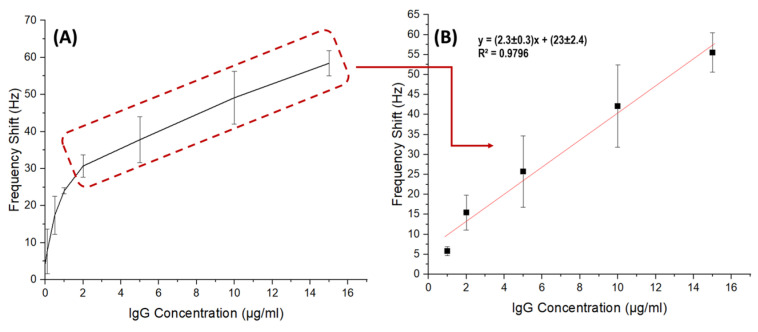
Analytical response (**A**), and calibration curve (**B**) of QTF immunosensor (*n* = 3).

**Figure 8 sensors-24-04319-f008:**
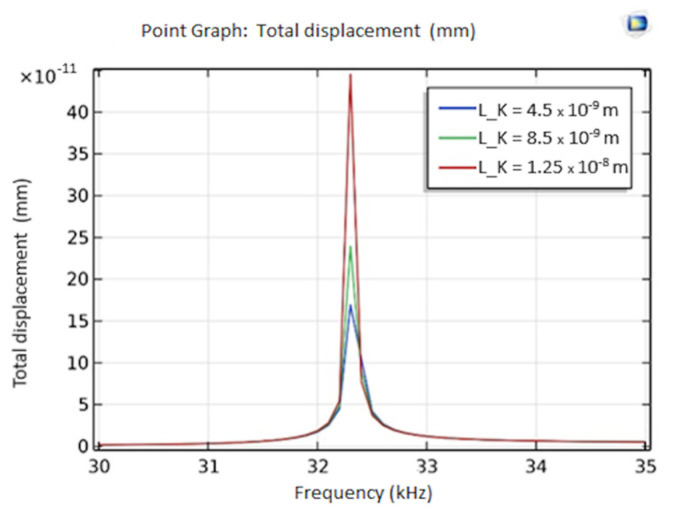
Resonance frequency response of QTF biosensor with respect to different amounts of interaction between anti-IgG and IgG including 4.5 nm, 8.5 nm, and 12.5 nm.

## Data Availability

The authors do not have permission to share data.

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
