# Peer review of "Anti-IgG Doped Melanin Nanoparticles Functionalized Quartz Tuning Fork Immunosensors for Immunoglobulin G Detection: In Vitro and In Silico Study"

_sensors, 2024, doi:10.3390/s24134319_

Round 1
Reviewer 1 Report
Comments and Suggestions for Authors
In this work, mass sensitive detection of IgG has been described using quartz tuning fork with immobilized anti-IgG antibodies via glutaraldehyde binding. The results are quite interesting but analytical performance of the immunosensor is challenging. The linear piece of the calibration plot contains only four experimental points and shows rather high deviation. The deep-and-dry technique used suffers from the humidity influence and is rather complex for practical use. The comparison of the results with other immunosensors based on traditional QCM sensors is necessary. The concentrations determined are rather high - what is potential application area of the immunosensor developed?
Comments on the Quality of English LanguageEnglish is quite satisfactory but needs minor changes (misprints and bulky sentences).
Author Response
Reviewers’ comments and author’s response
Referee 1
Comment 1: In this work, mass sensitive detection of IgG has been described using quartz tuning fork with immobilized anti-IgG antibodies via glutaraldehyde binding. The results are quite interesting but analytical performance of the immunosensor is challenging. The linear piece of the calibration plot contains only four experimental points and shows rather high deviation.
Response 1: Normally, we examined seven different concentrations to assess the analytical performance of the QTF-based immunosensor. However, only four of these concentrations fell within the linear range of the immunosensor. Due to budget constraints, we were unable to extend the experimental component of the research. Nevertheless, our research project on the QTF-based immunosensor has been accepted, and the pertinent information is acknowledged in the acknowledgments section. We are conducting further experiments for ovarian cancer diagnosis, using both in vitro and serum samples collected from healthy individuals and ovarian cancer patients. However, as the research is ongoing, we do not have any additional results to include in this manuscript at this time.
The referee raised concerns regarding the deviation in the resonance frequency shifts of the QTFs caused by the dip-and-dry technique. During surface modification, we utilized the QTF controller’s step motor for the immersion process. The immersion process's working principle depends on the distance between the liquid surface and the prongs. After immersion, the motor moves 0.4 μm, but this distance directly affects the dipping depth. Consequently, this situation caused deviations between measurements. We are working to enhance the controller to eliminate this issue.
Comment 2: The deep-and-dry technique used suffers from the humidity influence and is rather complex for practical use.
Response 2: The deep-and-dry technique suffers from not only the humidity. But it is a quite simple and versatile technique that enables us to coat irregular objects like quartz tuning fork. We had similar concerns with the reviewer, therefore, we have been tested different coating techniques such plasma polymerization, evaporation induced coating technique and electropolymerization for the QTF-based biosensor (Can et al., 2018; Özgüzar et al., 2019; Kaleli-Can et al., 2021; Demir et al., 2021; Can, 2024). We have seen that the stability and repoducibility of plasma polymerized thin film is problematic and the processability of evaporation induced coating technique and electropolymerization is difficult due to the small size and geometry of QTF. The QTF controller system has a platform that move 0,4 mm in every step. With the help of this controllable platform, we have obtain quite reproducible thin films compared to mentioned method. Therefore, we prefer to use deep-and-dry technique.
Can, G. K., Özgüzar, H. F., Kabay, G., Kömürcü, P., & Mutlu, M. (2018). Simultaneous insulation and modification of quartz tuning fork surface by single-step plasma polymerization technique with amine-rich precursors. MRS Communications, 8(2), 541-549.
Özgüzar, H. F., Can, G. K., Kabay, G., & Mutlu, M. (2019). Quartz tuning fork as a mass sensitive biosensor platform with a bi-layer film modification via plasma polymerization. MRS Communications, 9(2), 710-718.
Kaleli-Can, G., Özgüzar, H. F., & Mutlu, M. (2021). Development of mass sensitive sensor platform based on plasma polymerization technique: Quartz tuning fork as transducer. Applied Surface Science, 540, 148360.
Demir, D., GündoÄŸdu, S., Gundogdu, S., Kılıç, Åž., Kilic, S., KartallıoÄŸlu, T., ... & Kaleli Can, G. (2021). A comparison of different strategies for the modification of quartz tuning forks based mass sensitive sensors using natural melanin nanoparticles. Akıllı Sistemler ve Uygulamaları Dergisi, 4(2), 128-132.
Can, G. K. (2024). Molecularly Imprinted Chitosan Modified Quartz Tuning Fork Sensors for Real Time Biosensing in Liquid Environment. Düzce Üniversitesi Bilim ve Teknoloji Dergisi, 12(1), 337-347.
Comment 3: The comparison of the results with other immunosensors based on traditional QCM sensors is necessary.
Response 3: In comparing Quartz Crystal Microbalance (QCM) and Quartz Tuning Fork (QTF) biosensors for IgG detection, it is essential to consider their operational principles, sensitivity, and surface modification steps. QCM biosensors are widely recognized for their high sensitivity and rapid detection capabilities, making them attractive for early infection detection and disease monitoring. On the other hand, QTF biosensors, such as those used in QTF-based biosensors, offer unique advantages in detecting mass changes with high precision, showcasing their potential for specialized applications. However, due to the surface area, geometry and surface modification differences, limited the comparison of this two biosensor for too many variable.
Comment 4: The concentrations determined are rather high - what is potential application area of the immunosensor developed?
Response 4: In agreement with the reviewer's comments, the concentrations that were studied are indeed high given the current circumstances. QTF-based immunosensors are still a relatively new area and require further research to ultimately enhance their performance. Since our journey began in 2018 to develop QTF-based biosensors, we have tested various coating techniques to enhance sensor performance, including plasma polymerization, molecular imprinting, electroplating, dip coating, and evaporation-induced coating.
Nowadays, with the support of TUBÄ°TAK, we are working to further enhance the performance of these sensors and have collected promising results. However, this study is ongoing, and we have not yet been able to share the results in the literature. But with this performance, the potential application areas of the developed QTF-based immunosensor include the detection and monitoring of biomarkers in various fields such as medical diagnostics, environmental monitoring, and food safety.
Comment 5: English is quite satisfactory but needs minor changes (misprints and bulky sentences).
Response 5: We have improved the English in the revised version of manuscript.
Reviewer 2 Report
Comments and Suggestions for Authors
The general idea of using quartz-crystal tuning forks as an oscillator is interesting. After biological functionalization with anti-IgG antibodies, the target molecules can be detected in low concentration. However, I cannot recommend this manuscript for publication:
- The English writing is very poor, the manuscript would need corrections sentence by sentence.
- The references are not correctly numbered within the text.
- The introduction contains incorrect information regarding the comparison between QTF and QCM devices. It seems that the authors may not know this research field well.
- A motivation on "why to detect IgG" is missing. There are significant articles on immunosensors available in literature, including the IgG - anti-IgG binding. The reference list may be somehow incomplete.
- The Tables 1, 2, 3 can be moved to the Supporting Information.
- The steps of functionalizing the QTF devices with anti-IgG should best be explained in a schematic figure.
- The actual bioassay is not addressed (only 3 lines on page in Section 2.1.3): What was the liquid /buffer? Temperature? Did you measure in a biologically relevant liquid? Will the QTF device function with e.g. blood serum? Did you perform a negative control to proof that there is a molecular recognition effect? Did you check for non-specific adsorption? Potential readers, who are into the field of biosensors, will find this type of information highly relevant.
- Several figures have a poor quality and look like screen prints.
Comments on the Quality of English Language
The English writing is poor, this is not acceptable.
Author Response
Reviewers’ comments and author’s response
Reviewer 2
The general idea of using quartz-crystal tuning forks as an oscillator is interesting. After biological functionalization with anti-IgG antibodies, the target molecules can be detected in low concentration. However, I cannot recommend this manuscript for publication:
Comment 1: The English writing is very poor, the manuscript would need corrections sentence by sentence.
Response 1: We improved the english of the manuscript in the revised version.
Comment 2: The references are not correctly numbered within the text.
Response 2: In agreement with the reviewer's comments, the references were corrected.
Comment 3: The introduction contains incorrect information regarding the comparison between QTF and QCM devices. It seems that the authors may not know this research field well.
Response 3: We revised the comparison between QTF and QCM devices part of the manuscript:
“Among these approaches, quartz crystal microbalance (QCM) biosensors are one of the most preferred mass-sensitive biosensors and have been extensively studied for the de-tection of IgG by monitoring frequency shifts following IgG binding to a crystal [19–22]. Compared to QCM-based biosensors, quartz tuning fork (QTF) biosensors possess distinct advantages, including high quality factor, sensitivity, repeatability, frequency stability, sharp frequency response, and low cost [23–32]. These sensors can detect biomolecules in liquid environments by converting the mass of the target analytes into a signal, repre-sented as the resonance frequency of the QTF using an oscillator circuit and frequency counter [28,32–39]. The performance of the sensor is influenced by the interaction between the target biomolecule and the surface of the resonator. The modification of the trans-ducer’s surface through the creation of a recognition layer is essential due to the chemical inertness of the transducer’s surface, enhancing the sensitivity and selectivity of mass-sensitive biosensors for biomolecule detection [40]. Due to their small size, surface modifications of QTFs are challenging and require the use of specialized equipment, materials, reagents, and numerous repetitive laboratory tests. To address these chal-lenges, several simulation-based studies have been conducted to minimize optimization time and reduce the overall cost of the experimental tests. Atabaki et al. developed a finite element method (FEM)-based model using the COMSOL Multiphysics simulation pro-gram for early detection of acute myocardial infarction [41]. Another study simulated the response of polystyrene and polymethylmethacrylate modified QTF surfaces to volatile organic compounds (VOCs) with 97 % of accuracy [33]. Additionally, Sampson et al. de-signed graphene nanoribbons, polyaniline and gold nanoparticles capped with polyvinyl pyrrolidone modified QTF biosensor to identify VOCs such as ethanol, methanol, chlo-roform and acetone by using the FEM using COMSOL Multiphysics [34]. A correlation was found between the simulation results and the experimental results. The impressive predictive accuracy of simulations in biosensors opens an exciting opportunity to improve the effectiveness of QTF-based biosensors in biomedical applications via the exploration of innovative nanomaterials through simulation studies.”
Comment 4: A motivation on "why to detect IgG" is missing. There are significant articles on immunosensors available in literature, including the IgG - anti-IgG binding. The reference list may be somehow incomplete.
Response 4: We added related information to manuscript and extended the reference list:
“Immunoglobulin G (IgG) play a pivotal role in the immune response, serving as a primary defense mechanism against pathogens [1,2]. Detecting IgG levels can provide valuable information about an individual’s immune status, exposure to infections, and potential immunity against specific diseases [3–5]. Regarding infectious diseases such as COVID-19, the detection of IgG plays a key role in understanding the immune response post-infection and assessing the development of immunity over time [6]. Recent studies indicate that the presence of anti-SARS-CoV-2 IgG can indicate past exposure to the virus and potentially confer protection against reinfection [3]. As a result of monitoring the levels of IgG in COVID-19 patients as well as in healthy individuals, it can be possible to ascertain the duration of immunity and to guide public health strategies, including vaccination campaigns and infection control measures [6]. IgG are also vital for diagnosing and treating autoimmune disorders and thrombotic conditions. A significant proportion of patients with thrombotic microangiopathies and severe ADAMTS13 deficiency have an-ti-ADAMTS13 IgG, highlighting the usefulness of IgG detection in these clinical settings [7]. The identification of anti-mitochondrial IgG antibodies has been linked to specific signs and symptoms of systemic lupus erythematosus, emphasizing the importance of detecting IgG antibodies for understanding the pathogenesis of the disease [8]. As a result of the advancement of biosensor technologies, sensitive and rapid methods of detecting IgG have been developed, which are facilitating point-of-care testing and personalized medicine practices [9]. In the last few decades, innovative biosensors have revolutionized diagnostic capabilities for IgG detection, enabling early detection of diseases and mon-itoring of therapeutic outcomes. In addition to standard analytical methods, such as the enzyme-linked immunosorbent assay (ELISA), there are several biosensor designs that use chemical, electrochemical, optical, and mass sensitive techniques to measure IgG levels in blood and plasma [10–18]. Among these approaches, quartz crystal microbalance (QCM) biosensors are one of the most preferred mass-sensitive biosensors and have been extensively studied for the detection of IgG by monitoring frequency shifts following IgG binding to a crystal [19–22]. Compared to QCM-based biosensors, quartz tuning fork (QTF) biosensors possess distinct advantages, including high quality factor, sensitivity, repeatability, frequency stability, sharp frequency response, and low cost [23–32]. These sensors can detect biomolecules in liquid environments by converting the mass of the target analytes into a signal, represented as the resonance frequency of the QTF using an oscillator circuit and frequency counter [28,32–39]. The performance of the sensor is in-fluenced by the interaction between the target biomolecule and the surface of the reso-nator. The modification of the transducer’s surface through the creation of a recognition layer is essential due to the chemical inertness of the transducer’s surface, enhancing the sensitivity and selectivity of mass-sensitive biosensors for biomolecule detection [40]. Due to their small size, surface modifications of QTFs are challenging and require the use of specialized equipment, materials, reagents, and numerous repetitive laboratory tests. To address these challenges, several simulation-based studies have been conducted to minimize optimization time and reduce the overall cost of the experimental tests. Atabaki et al. developed a finite element method (FEM)-based model using the COMSOL Multiphysics simulation program for early detection of acute myocardial infarction [41]. Another study simulated the response of polystyrene and polymethylmethacrylate modified QTF surfaces to volatile organic compounds (VOCs) with 97 % of accuracy [33]. Additionally, Sampson et al. designed graphene nanoribbons, polyaniline and gold nanoparticles capped with polyvinyl pyrrolidone modified QTF biosensor to identify VOCs such as ethanol, methanol, chloroform and acetone by using the FEM using COMSOL Multiphysics [34]. A correlation was found between the simulation results and the experimental results. The impressive predictive accuracy of simulations in biosensors opens an exciting opportunity to improve the effectiveness of QTF-based biosensors in biomedical applications via the exploration of innovative nanomaterials through simu-lation studies.”
Comment 5: The Tables 1, 2, 3 can be moved to the Supporting Information.
Response 5: We have moved the tables 1, 2, and 3 to the supporting information.
Comment 6: The steps of functionalizing the QTF devices with anti-IgG should best be explained in a schematic figure.
Response 6: We draw a schematic figure and added to revised version of the manuscript.
Comment 7: The actual bioassay is not addressed (only 3 lines on page in Section 2.1.3): What was the liquid /buffer? Temperature? Did you measure in a biologically relevant liquid? Will the QTF device function with e.g. blood serum? Did you perform a negative control to proof that there is a molecular recognition effect? Did you check for non-specific adsorption? Potential readers, who are into the field of biosensors, will find this type of information highly relevant.
Response 7:
Thank you for your thorough review and insightful comments. We appreciate the opportunity to address the concerns you raised regarding the details of the bioassay and the performance of the QTF-based immunosensor. In this study, we were tried to simulate the in vitro experiment by using FEM in COMSOL program. Therefore, we, firstly, focused on the development of immunosensor in vitro. Firstly, we acknowledge that the bioassay section is indeed brief. To provide a more comprehensive understanding, we have expanded the section. In our experiments, the bioassay was conducted using a phosphate-buffered saline (PBS) solution with a pH of 7.4. The temperature was maintained at 25°C to ensure optimal conditions for biological reactions. The related information was added to the revised version of the manuscript. While this study utilized PBS, we recognize the importance of testing biologically relevant liquids. Indeed, ongoing experiments involve measurements in complex matrices such as blood serum to evaluate sensor performance under more realistic conditions. We are currently conducting experiments with biologically relevant liquids, including blood serum, to further validate the sensor's performance supported by TUBÄ°TAK. In each experiment, we used unmodified QTFs as a negative control. But due to QTFs' inert nature, no frequency change was observed. We hope these clarifications and additional details address the concerns raised. We are committed to advancing our research and sharing our findings with the scientific community as our studies progress. Thank you again for your valuable feedback.
Comment 8: Several figures have a poor quality and look like screen prints.
Response 8: We improved the quality of the figures in the revised version of manuscript.
Round 2
Reviewer 1 Report
Comments and Suggestions for Authors
I agree with most of hte amendments and changes made but the problem of analytical perfromance is still far from solution. iff the authors did not have possibility to increase the number of experimrnts within linear arange of concentration curve, they should mention that the results presented are only firsts steps toward analytical applications. Section 3.4 contains calibration curve with exhausting number of significant digits, which should be reduced, stadard deviation of the slope and intercept should be added (line 340).
Author Response
Comment 1: I agree with most of hte amendments and changes made but the problem of analytical perfromance is still far from solution. iff the authors did not have possibility to increase the number of experimrnts within linear arange of concentration curve, they should mention that the results presented are only firsts steps toward analytical applications.
Response 1: We added information about the limitation of the research as follows:
“The present manuscript presents preliminary findings. These initial results are foundational and form the first steps toward developing robust analytical applications for our QTF-based biosensor. Further research will focus on expanding the range of concentrations tested and optimizing the sensor's performance to enhance its analytical capabilities.”
Comment 2: Section 3.4 contains calibration curve with exhausting number of significant digits, which should be reduced, standard deviation of the slope and intercept should be added (line 340).
Response 2: Page 12, Line 354: We added std deviation of the slope and intercept and reduce the significant digits to Figure 6(B) in the revised version of manuscript.

Reviewer 2 Report
Comments and Suggestions for Authors
The authors did an in-depth revision of their manuscript and I feel that it can almost be published. Severla aspects should still be improved:
- In Figure 2, I would expect that the signals will saturate with time, which is not the case when looking at the data. All readers, who are active in the field of biosensors, will have this question. Maybe the authors did not wait long enough until the signals were really in equilibrium? I would repeat this experiment and simply wait longer in between the subsequent steps of the functionalization protocol.
- Furthermore, I wonder why the resonance frequency seems to show first a decrease of the resonance frequency, and then an increase of the frequency. This pattern repeats during each functionalization step. Based on experience (and theory, see Sauerbrey equation), I would expect that each additional layer of functionalization causes an additional frequency drop. So, it will be impossible that the resonator has the same frequency after functionalization as before.
- The legend at the x-axis is "time in units of second to the power of minus 3". Why is the time not simply given in seconds, or minutes?
- If possible, please provide and English translation for Reference [53]. Then, you can still mention that the original reference is in Turkish language.
Author Response
Referee 2
The authors did an in-depth revision of their manuscript and I feel that it can almost be published. Several aspects should still be improved:
Comment 1: In Figure 2, I would expect that the signals will saturate with time, which is not the case when looking at the data. All readers, who are active in the field of biosensors, will have this question. Maybe the authors did not wait long enough until the signals were really in equilibrium? I would repeat this experiment and simply wait longer in between the subsequent steps of the functionalization protocol. Furthermore, I wonder why the resonance frequency seems to show first a decrease of the resonance frequency, and then an increase of the frequency. This pattern repeats during each functionalization step. Based on experience (and theory, see Sauerbrey equation), I would expect that each additional layer of functionalization causes an additional frequency drop. So, it will be impossible that the resonator has the same frequency after functionalization as before.
Response 1: QTF-based biosensors work on a similar principle to QCM, but the controller that is used for both surface modification and measurement works under the different forces. Using the QTF-F controller, the frequency shift (Δf) of the biosensor in solution was monitored by recording the measurements every 103 seconds, corresponding to a frequency resolution of 0.00001 Hz, during each immersion conducted for surface modification and measurement purposes. The frequency variation in the QTF biosensor positioned in the liquid results from the simultaneous influence of multiple forces. The observed frequency decrease immediately after immersion and its subsequent rise over time can be attributed to the capillary effect of the liquid between the prongs, the greater damping effect of the liquid phase compared to air, the immobilization of MNP, GLU, and anti-IgG on the surface, the binding of IgG to anti-IgG decorated melanin nanoparticles, and temperature-dependent evaporation. Therefore, it is crucial that we observe these effects during time-dependent frequency measurements and analyze the moment when the frequency change is recorded. To achieve the objectives of this research, steps have been taken to modify QTFs that promote piezoelectricity. To perform the experiment, the first step was to remove the crystal forks from their casings, which vibrate at a frequency of 32768 Hz under vacuum, by breaking the vacuum seal in a laboratory environment. Afterwards, the forks were cleaned with a sonicator and dried with nitrogen gas. Afterward, the damping effect in air was measured for each sample individually, establishing baseline values. Although QTFs is constant at a certain frequency when they are in a vacuum, once the vacuum is broken, they can vibrate at a different resonance frequency. An analysis of preliminary results shows that the average frequency was found as 32759.1 ± 0.6 Hz (n=30) (Can et al., 2018; Özgüzar et al., 2019; Kaleli-Can, 2018; Kaleli-Can et al., 2020; Kaleli-Can et al., 2021; Demir et al., 2021; Kaleli-Can et al., 2022). Forks that deviated from these specified values were excluded from the next stage.
As shown in Figure 2, when the sensor is immersed in solution, a sharp frequency drop is observed due to the damping effect and capillary effect of the liquid at the moment of immersion. However, owing to the QTF geometry, equilibrium is rapidly achieved, resulting in frequency stabilization. MNPs adhesion to the surface is monitored for one hour following immersion. During this process, only a maximum of 40 µm of the prong is in contact with the solution. Over time, the frequency increases due to surface evaporation. We added a video during the submission about the relation between MNP solution and prongs of the QTFs, and it can be observed the mentioned behavior from this video, clearly. By subtracting the frequency value before evaporation onset from the equilibrium frequency value, real-time monitoring of surface adhesion is enabled.
After the surface coating with MNPs, the prongs are washed to remove unbound nanoparticles and then allowed to dry. Frequency measurements taken in air before and after the modification steps are plotted against time to show the period of contact with the IgG solution and the prong surface. The real-time observation of changes in the QTF-based biosensor following each modification and subsequent IgG capture is determined by subtracting the frequency value before evaporation from the equilibrium frequency value. The prongs are washed to prevent non-specific binding and allowed to dry. The frequency value after drying is subtracted from the dry phase frequency value before immersion, as shown in Figure 2, to calculate the frequency change in the air.
Related information was added to manuscript:
Page 6, Line 240: “QTF-based biosensors work on a similar principle to QCM, but the controller used for both surface modification and measurement operates under different forces. Using the QTF-F controller, the frequency shift (Δf) of the biosensor in solution was monitored by recording measurements every 103 seconds, corresponding to a frequency resolution of 0.00001 Hz, during each immersion conducted for surface modification and measurement purposes. The frequency variation in the QTF biosensor positioned in the liquid results from the simultaneous influence of multiple forces. The observed frequency decrease immediately after immersion and its subsequent rise over time can be attributed to the capillary effect of the liquid between the prongs, the greater damping effect of the liquid phase compared to air, the immobilization of MNP, GLU, and anti-IgG on the surface, the binding of IgG to anti-IgG decorated melanin nanoparticles, and temperature-dependent evaporation. Therefore, it is crucial to observe these effects during time-dependent frequency measurements and analyze the moment when the frequency change is recorded.
As shown in Figure 2, when the sensor is immersed in solution, a sharp frequency drop is observed due to the damping effect and capillary effect of the liquid with the immersion. However, owing to the QTF geometry, equilibrium is rapidly achieved, resulting in frequency stabilization. MNPs adhesion to the surface is monitored for one hour following immersion. During this process, only a maximum of 40 µm of the prong is in contact with the solution. Over time, the frequency increases due to surface evaporation.”
Comment 2: The legend at the x-axis is "time in units of second to the power of minus 3". Why is the time not simply given in seconds, or minutes?
Response 2: Page 7, Line 233: The x axis labed of Figure 1 was changed to millisecond (ms) in the revised version of the manuscript.
Comment 3: If possible, please provide and English translation for Reference [53]. Then, you can still mention that the original reference is in Turkish language.
Response 3: We revised and added the English translation for Reference [53] as follows:
“Akman, B.; Ä°slam, B.; Kaleli Can G.; TopaloÄŸlu AvÅŸar, N.; Åžen Karaman, D.; Baysoy, E. As an Alternative Photocatalyst Under UV-A Irradiation for Food and Health Applications: Natural Melanin Nanoparticles. European Journal of Science and Technology 2021, 940–946, (published in Turkish language) doi:10.31590/ejosat.1040830.”

Round 3
Reviewer 2 Report
Comments and Suggestions for Authors
Dear authors,
Now, I can understand that the tuning forks were only dipped by a small part into the liquid under study. This explains a lot.
However, the newly added text is not clear at one point, see page 6, lines 249 - 250:
You mention that measurements were recorded every 1000 seconds and that this corresponds to a frequency resolution of 0.00001 Hz. One measurement within 1000 seconds should correspond to 0.001 Hz. Furthermore, when looking at Figure 2, I assume that data points are recorded at least once per second, once in 1000 milli seconds.
Please understand that reader will not understand this, it should be corrected - or, if it is correct, explained better.
Author Response
Comment 1:
Dear authors,
Now, I can understand that the tuning forks were only dipped by a small part into the liquid under study. This explains a lot.
However, the newly added text is not clear at one point, see page 6, lines 249 - 250:
You mention that measurements were recorded every 1000 seconds and that this corresponds to a frequency resolution of 0.00001 Hz. One measurement within 1000 seconds should correspond to 0.001 Hz. Furthermore, when looking at Figure 2, I assume that data points are recorded at least once per second, once in 1000 milli seconds.
Please understand that reader will not understand this, it should be corrected - or, if it is correct, explained better.
Response 1:
We appreciate the time and effort the reviewers have dedicated to providing valuable feedback on our manuscript. With their comments, the manuscript has reached a better version and become clearer for the reader. We are aware that QTF-based biosensors are a relatively new area, and additionally, we are using a different type of controller compared to other studies. Therefore, these revisions have significantly improved the manuscript. After receiving the referee's comments, we contacted Asensis, the company that produces the controller, to clarify this point. They indicated that the device records ten frequency measurements per second, with a frequency resolution of 0.01 Hz. Therefore, we corrected that part in the manuscript:
“Using the Asensis QTF F-master, the frequency shift (Δf) of the biosensor in solution was monitored by recording ten frequency measurements per second with a frequency resolution of 0.01 Hz.”
